# Trends and Advances in Wearable Plasmonic Sensors Utilizing Surface-Enhanced Raman Spectroscopy (SERS): A Comprehensive Review

**DOI:** 10.3390/s25051367

**Published:** 2025-02-23

**Authors:** Svetlana N. Khonina, Nikolay L. Kazanskiy

**Affiliations:** 1Samara National Research University, 34 Moskovskoye Shosse, Samara 443086, Russia; kazanskiy@ipsiras.ru; 2Image Processing Systems Institute, NRC “Kurchatov Institute”, 151 Molodogvardeyskaya, Samara 443001, Russia

**Keywords:** wearable plasmonic sensors, surface plasmon resonance, localized surface plasmon resonance, surface-enhanced Raman spectroscopy, flexible substrate

## Abstract

Wearable sensors have appeared as a promising solution for real-time, non-invasive monitoring in diverse fields, including healthcare, environmental sensing, and wearable electronics. Surface-enhanced Raman spectroscopy (SERS)-based sensors leverage the unique properties of SERS, such as plasmonic signal enhancement, high molecular specificity, and the potential for single-molecule detection, to detect and identify a wide range of analytes with ultra-high sensitivity and molecular selectivity. However, it is important to note that wearable sensors utilize various sensing mechanisms, and not all rely on SERS technology, as their design depends on the specific application. This comprehensive review highlights the recent trends and advancements in wearable plasmonic sensing technologies, focusing on their design, fabrication, and integration into practical wearable devices. Key innovations in material selection, such as the use of nanomaterials and flexible substrates, have significantly enhanced sensor performance and wearability. Moreover, we discuss challenges such as miniaturization, power consumption, and long-term stability, along with potential solutions to address these issues. Finally, the outlook for wearable plasmonic sensing technologies is presented, emphasizing the need for interdisciplinary research to drive the next generation of smart wearables capable of real-time health diagnostics, environmental monitoring, and beyond.

## 1. Introduction

Wearable sensors for health monitoring are revolutionizing modern healthcare by combining advancements in materials science, data analytics, and wireless communication [1,2]. These innovative devices enable continuous tracking of vital physiological parameters, incorporating heart rate, blood oxygen levels, skin temperature, and activity patterns, delivering real-time perceptions into an individual’s health [3,4,5]. Recent developments in stretchable and flexible electronics allow these sensors to integrate seamlessly into clothing or adhere directly to the skin for unobtrusive and comfortable use [6,7,8,9]. Enhanced by artificial intelligence (AI), they can identify early signs of disease, customize health interventions, and predict critical events like arrhythmias or hypoglycemia [10]. Moreover, smart contact lenses are innovative wearable devices that integrate advanced technology into contact lenses or eyeglasses to provide a range of functionalities beyond vision correction [11,12]. These lenses often incorporate sensors, microelectronics, and wireless communication capabilities to monitor health metrics such as intraocular pressure, glucose levels, or environmental factors like UV exposure [12,13]. Beyond personal health, aggregated anonymized data from wearables offer valuable insights for public health and epidemiological studies. While challenges such as ensuring accuracy, optimizing energy efficiency, and safeguarding privacy persist, advancements in device miniaturization, battery design, and secure cloud technologies are steadily overcoming these barriers, heralding a new era of preventive and personalized medicine [14,15].

Plasmonics, which studies the interaction between light and free electrons in metallic nanostructures, plays a crucial role in wearable sensors [16,17,18]. Two primary mechanisms are leveraged: localized surface plasmon resonance (LSPR) and surface-enhanced Raman spectroscopy (SERS). While both involve plasmonic effects, their sensing principles differ significantly. LSPR sensors measure shifts in resonance wavelength due to changes in the local refractive index, making them highly effective for detecting molecular binding events [19]. In contrast, SERS sensors amplify Raman scattering signals, allowing for direct molecular identification with ultra-high sensitivity [20]. Given the scope of this review, we focus specifically on SERS-based wearable sensors, which utilize plasmonic nanostructures to enhance Raman signals for biomarker detection in sweat and other biofluids.

Sweat is an ideal biofluid for use in wearable plasmonic sensors because of its non-invasive collection and rich composition of biomarkers [21,22,23]. Unlike blood, sweat can be continuously sampled without the need for needles, enhancing user comfort and enabling real-time health monitoring [24]. It contains a wide range of analytes, including electrolytes, metabolites, hormones, and proteins, which are critical indicators of physiological states and diseases. Wearable plasmonic sensors leverage the interaction of light with nanoscale materials to achieve high sensitivity in detecting these biomarkers [25]. Additionally, the compatibility of sweat with advanced nanomaterials, such as gold (Au) and silver (Ag) nanoparticles, enhances the plasmonic effects, enabling precise and rapid detection. This makes sweat-based wearable plasmonic sensors an excellent tool for personalized medicine, fitness tracking, and early disease detection [26,27,28]. Figure 1 presents the characteristics of sweat, which is particularly suitable for sensing biomarkers for health monitoring in wearable sensors.

Innovations in nanofabrication have empowered the integration of plasmonic nanostructures into flexible substrates, granting seamless incorporation into wearable formats like skin patches or textiles [29]. Moreover, plasmonic sensors can be coupled with advanced optical systems and machine learning (ML) algorithms to augment specificity and interpret complex data patterns. The ultrafast response times and potential for multiplexed detection make plasmonics a game-changer for applications ranging from chronic disease management to fitness monitoring. As researchers address challenges in scalability, durability, and power efficiency, wearable plasmonic sensors hold the promise of delivering precise, non-invasive health insights, bridging the gap between advanced diagnostics and everyday wellness monitoring [20]. There are various sensing techniques available for detecting biomarkers, including colorimetry, surface-enhanced Raman spectroscopy (SERS), fluorescence, and electrochemiluminescence. Each of these methods has its own unique advantages, depending on the application. However, in this review, we have focused specifically on wearable plasmonic sweat sensors that utilize the SERS technique. This method has gained significant attention due to its high sensitivity and ability to detect specific biomarkers in sweat, making it a favorable tool for non-invasive health monitoring.

**Figure 1 sensors-25-01367-f001:**
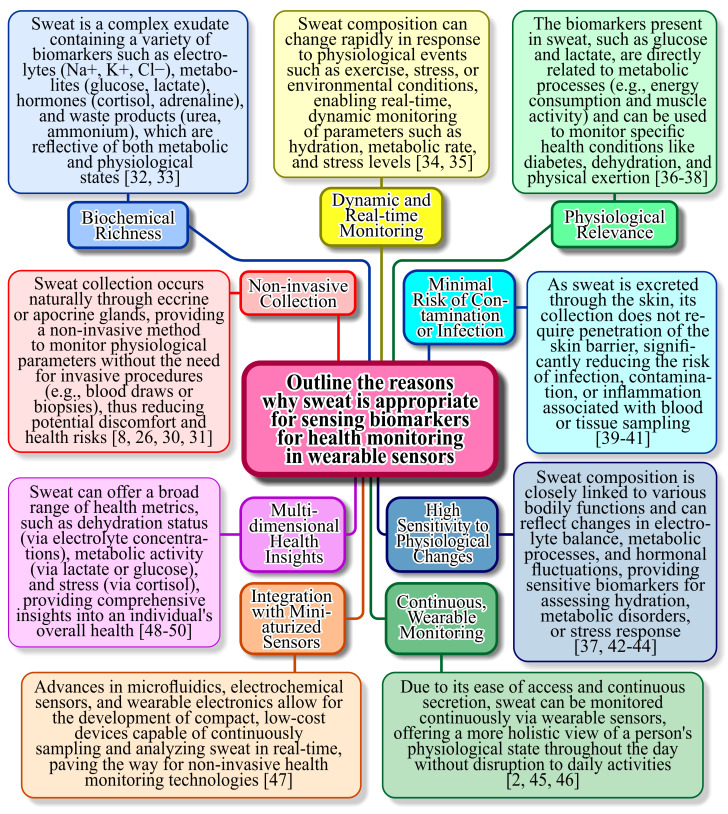
Reasons and scientific explanation why sweat is particularly suitable for sensing biomarkers for health monitoring in wearable sensors [2,8,26,30,31,32,33,34,35,36,37,38,39,40,41,42,43,44,45,46,47,48,49,50].

## 2. Fundamentals of Plasmonics

In this section, we delved into the fundamental principles of plasmonics, explored the diverse range of materials utilized in plasmonic applications, and highlighted the critical design considerations essential for the expansion of high-performance wearable sensors. By examining the interplay between plasmonic phenomena and material properties, we provided insights into how these elements synergize to achieve enhanced sensitivity and functionality. Additionally, we discussed key factors such as flexibility, biocompatibility, and integration strategies, which are pivotal for optimizing wearable plasmonic sensors for real-world applications.

### 2.1. Plasmonic Principles

SERS is a powerful analytical technique that combines surface-enhanced Raman scattering with resonance effects to achieve highly sensitive molecular detection [51]. SERS amplifies Raman signals by leveraging the interaction of analyte molecules with metallic nanostructures, which produce localized EM field enhancements (Figure 2) [52,53]. Unlike SPR and LSPR, which primarily monitor changes in RI or optical properties near the sensor surface, SERS provides direct molecular identification through vibrational spectroscopy, offering chemical specificity. Additionally, the resonant enhancement in SERS further amplifies the signal for analytes that absorb light at the laser excitation wavelength, significantly improving sensitivity [54]. This dual enhancement—structural and resonance-based—makes SERS particularly advantageous for detecting trace amounts of analytes and studying molecular interactions with unparalleled specificity and sensitivity compared to SPR and LSPR [55,56].

LSPR occurs in metallic nanostructures when conduction electrons collectively oscillate in response to incident light, creating intense localized electromagnetic fields. This phenomenon underpins both LSPR-based refractive index sensors and SERS sensors, but their sensing strategies are distinct. LSPR sensors detect analytes by monitoring shifts in resonance wavelength due to RI changes [57,58,59]. SERS sensors, on the other hand, leverage these localized plasmonic fields to amplify Raman scattering signals, enhancing molecular fingerprint detection. While both rely on plasmonic enhancement, LSPR sensors are primarily used for continuous monitoring of environmental changes, whereas SERS sensors provide highly specific molecular identification. This distinction is critical when considering optimization strategies, as design considerations for LSPR do not necessarily apply to SERS-based sensors in wearable applications.

### 2.2. Materials for Plasmonics

The choice of materials significantly influences the performance of plasmonic sensors [60,61,62,63,64]. Traditionally, noble metals such as Au and Ag have been the materials of choice due to their excellent plasmonic properties, including high conductivity and stability. Au is favored for its chemical inertness and biocompatibility, making it ideal for biosensing applications [65]. Ag offers superior plasmonic performance in terms of sharper resonances but suffers from susceptibility to oxidation, limiting its long-term durability [66].

Emerging materials are expanding the horizons of plasmonics [67]. Aluminum, for instance, is gaining attention due to its abundance, cost-effectiveness, and suitability for applications in the ultraviolet spectrum [68]. Graphene, a two-dimensional carbon material, has introduced tunable plasmonic properties and flexibility, enhancing its integration into wearable devices [69]. Hybrid materials, combining plasmonic metals with semiconductors or dielectric materials, are being explored to achieve improved performance metrics, such as enhanced sensitivity and broader spectral tenability [70]. These advances are critical for adapting plasmonics to the diverse requirements of wearable sensors.

Despite their potential, scalable, cost-effective, and eco-friendly methods to fabricate high-performance SERS and SEPL substrates with broad plasmonic response and significant enhancement remain elusive [71,72,73]. Kuchmizhak et al. introduced single-pulse laser-induced spallative micron-scale craters in thick silver films, featuring internal nanoscale tips, as efficient substrates with strong polarization-dependent SEPL and SERS enhancements [74]. Using a nanometer-thick Rhodamine 6G layer, the substrates achieved enhancement factors of 40 for SEPL and 2 × 10^6^ for SERS, marking a significant advancement in plasmonic substrate performance. Furthermore, the research provided a detailed investigation into the physical mechanisms behind the formation of these nanotextures in thick metal films. Ablation behavior was shown to be hybrid and fluence-dependent: low fluences generated spallative craters characteristic of bulk materials, while higher fluences produced upright nanoscale tips (frozen nanojets) at crater centers, typically associated with thin-film ablation. This approach represents a promising, low-cost, and straightforward pathway for large-scale fabrication of efficient spallation-textured plasmonic substrates. Leveraging MHz-repetition-rate femtosecond fiber lasers with multiplexed beams, this method provides a practical solution for routine chemical and biological sensing applications [74].

Tightly focused femtosecond Laguerre–Gaussian pulses of visible laser radiation can be utilized for single-pulse ablation-based nanostructuring of a 50 nm gold film [75]. Lower-energy pulses of the same type are usually used to efficiently excite plasmonic photoluminescence in a rhodamine 6G dye monolayer deposited on the fabricated nanostructures. The symmetry between the beam’s shape and polarization at lower energies and the nanostructures produced at higher energies offers a promising approach for precise nanostructuring and enhanced surface spectroscopy using these tailored nanostructures [76].

Chowdhury et al. introduced a next-generation wearable SERS-active sensor that was stretchable, flexible, and optimized for single-molecule detection [77]. The device achieved an extraordinary SERS enhancement factor of approximately 10^11^, coupled with essential attributes for practical sensing, including a high scattering-to-absorption ratio (~2.5) and a large hotspot volume (40 nm × 40 nm × 5 nm). To enhance mechanical flexibility, biocompatible and transparent polydimethylsiloxane (PDMS) was utilized as the substrate, ensuring both durability and adaptability (Figure 3). Utilizing capacitive coupling in heart-shaped nanodimers with nanoscale gaps, the sensor suppressed plasmon resonance decay, achieving an exceptional SERS enhancement factor of 10^1^⁰ to 10^11^, facilitating single-molecule detection. Computational modeling confirmed that the sensor maintains strong SERS performance even when subjected to bending angles up to 100° and stretching up to 50%. Its simple fabrication process, outstanding flexibility, and ultra-high sensitivity position this sensor as a promising innovation for next-generation wearable devices, particularly in applications related to continuous health monitoring and personalized diagnostics [77].

Recently, Borodaenko et al. showcased the formation of self-organized laser-induced periodic surface structures (LIPSSs) on crystalline silicon wafers, achieving a record periodicity of 70 ± 10 nm through direct femtosecond laser ablation in isopropanol [78]. This nanoscale morphology arose from periodic surface patterning of photoexcited silicon via interference effects, followed by grating period reduction through Rayleigh–Taylor hydrodynamic instability. The resulting deep subwavelength LIPSSs exhibited exceptional anisotropic anti-reflection properties, enabling efficient coupling of incident far-field radiation into electromagnetic “hot spots” within silicon nanogaps. These features paved the way for optical biosensing platforms that capitalize on robust interactions between quantum emitters and confined light fields. The demonstrated 80-fold enhancement in spontaneous emission from an organic dye nanolayer and real-time optical monitoring of catalytic molecular transformations underscored the promise of bare and metal-coated subwavelength Si LIPSSs as versatile, cost-efficient biosensing platforms [78].

### 2.3. Key Design Considerations

Designing wearable plasmonic sensors involves balancing sensitivity, specificity, and durability. Sensitivity is a measure of how effectively the sensor detects changes in its environment, often determined by the quality factor of the plasmonic resonance [79]. Specificity ensures that the sensor selectively responds to the target analyte, which can be enhanced through functionalization of the plasmonic surface with receptor molecules or polymers [80]. Durability is crucial for wearable applications, as these sensors must withstand mechanical stress, exposure to sweat, and environmental conditions over extended periods [81]. This requires robust materials and protective coatings that do not compromise plasmonic performance [82]. Integration into wearable formats introduces additional challenges, such as maintaining flexibility and conformability without sacrificing optical properties. Figure 4 outlines the key features and challenges associated with wearable plasmonic sensors, providing a clear overview of their potential and limitations.

Flexible substrates, such as polymers and textiles, are often employed to address these requirements, while advancements in nanofabrication techniques enable precise deposition of plasmonic nanostructures on these substrates [89]. Together, these considerations drive the development of wearable plasmonic sensors that are both functional and practical for real-world applications [90]. Wang et al. introduced a wearable plasmonic-electronic sensor with the ability to recognize molecules universally (Figure 5a) [16]. Figure 5b illustrates the design of the wearable sensing device integrated with plasmonic metamaterials. Figure 5c displays optical images of the fabricated device. An enlarged view in Figure 5d reveals key components, including stretchable spiral fractal electrodes, an acetylcholine chloride-loaded hydrogel layer, and the NC metafilm. SEM images (Figure 5e,f) show the Ag NC metafilm as a densely packed, highly ordered array, confirming successful fabrication [16]. The sensor employed a flexible plasmonic metasurface with SERS capabilities, engineered to retain its plasmonic properties despite the mechanical stresses caused by bending and stretching. Combined with a flexible electronic system that extracted sweat non-invasively, the device analyzed analytes by capturing their unique SERS spectral signatures. As a proof of concept, the sensor successfully tracked trace levels of drugs in the body and provided insights into individual drug metabolism. By enabling precise and sensitive molecular monitoring, this innovation addressed key gaps in wearable technology and sets the stage for more advanced health assessments [16].

## 3. Fabrication and Integration Techniques

In this section, we provide a concise overview of the fabrication techniques, flexible substrate integration, as well as the powering and readout mechanisms of wearable plasmonic sensors.

### 3.1. Nanostructure Fabrication

#### 3.1.1. Top-Down Approaches (e.g., Lithography)

Top-down fabrication techniques, such as lithography, have been pivotal in defining nanostructures with precise geometries and features [91]. Lithographic methods, including photolithography [92] and electron-beam lithography (EBL) [93], allow the creation of nanoscale patterns with high resolution, making them ideal for plasmonic applications where control over nanostructure dimensions is critical for optical properties [20]. However, these methods often face challenges related to scalability and cost, especially for large-area production. Advanced lithography techniques, such as nanoimprint lithography (NIL) [94] and extreme ultraviolet (EUV) lithography [95], are emerging to address these issues, offering improved throughput and finer feature sizes while reducing production costs [96].

Colnita et al. presented the development of highly sensitive substrates for SERS detection platforms, utilizing NIL to fabricate nanotrenches in plastic substrates, which are subsequently coated with nanostructured Ag films [97]. These films, with thicknesses ranging from 10 nm to 100 nm, were deposited via direct current (DC) sputtering. By varying the deposition times, the Ag film thickness was systematically increased and its impact on the SERS enhancement factor was evaluated. The morphological and structural properties of the metalized nanotrenches were characterized using SEM and atomic force microscopy (AFM). To test the SERS activity, crystal violet (CV) was employed as an analyte on substrates with and without the nanoimprinted pattern. This approach integrated three critical factors for enhancing SERS performance: the metallization of low-cost, flexible substrates with Ag, the influence of Ag film thickness, and the periodic nanotrench pattern introduced by NIL. The results revealed a substantial enhancement in the SERS signal due to the periodic Ag nanopattern, with an enhancement factor (EF) reaching up to 10^7^ for a substrate featuring a 25 nm Ag layer atop the nanotrenches. The cumulative contribution of the plasmonic nanostructures within the Ag films, along with the periodic nanopatterned trenches, was carefully assessed, showcasing a synergistic enhancement effect [97].

Recently, Zeng et al. presented a simple fabrication method for silver-coated PMMA nanoparticles-on-a-mirror SERS substrates, where ultra-small metallic nanogaps and a reflective silver mirror amplify the electric field for stronger signals [98]. Optimizing the PMMA layer aligned the absorption peak with the excitation wavelength, maximizing intensity. The optimized substrate detected CV at 10^−13^ M under 532 nm laser excitation, with 7.75% RSD and stability for 37 days. This method offered a promising approach for high-performance SERS sensors in chemical and biochemical sensing.

#### 3.1.2. Bottom-Up Approaches (e.g., Self-Assembly)

Bottom-up approaches, such as molecular self-assembly and colloidal synthesis, rely on the intrinsic properties of materials to organize into nanostructures [99,100,101,102]. These methods offer significant advantages in scalability and material efficiency compared to top-down methods [103]. For instance, the self-assembly of nanoparticles can yield highly uniform and reproducible plasmonic structures with tunable optical properties by controlling parameters like particle size, shape, and interparticle spacing [104,105]. Hybrid approaches that combine top-down and bottom-up techniques are increasingly being explored to leverage the precision of lithography with the efficiency and cost-effectiveness of self-assembly [99].

In the study conducted by Mhlanga et al., SERS substrates were successfully fabricated on solid supports, including silicon wafers, gold-coated glass slides, and bare microscope slides [106]. These substrates featured LSPR metallic clusters immobilized on the supports. The fabrication process involved a two-step approach: first, the self-assembly of Ag or Au nanoparticles onto the substrates, followed by the in-situ deposition of a bulk Au layer. The bimetallic substrates exhibited significantly enhanced SERS performance compared to monometallic Au substrates, with enhancement factors of 2.5 × 10^7^ for Au-coated glass slides and 2.0 × 10^7^ for bare glass slides. Overall, the prepared substrates demonstrated strong SERS activity, highlighting their potential for high-sensitivity molecular detection [106].

Furthermore, Chen et al. presented a new class of SERS substrates—soft plasmene nanosheets—fabricated through a bottom-up self-assembly approach [107]. These ultrathin, flexible, and optically translucent nanosheets seamlessly conformed to complex real-world surfaces, including paper, plastic banknotes, and metallic coins, enabling highly reproducible and sensitive chemical detection. Their high transparency allowed for direct SERS signal acquisition, eliminating the need for additional substrates. By precisely controlling the size and shape of the nanosheet’s building blocks, both excitation wavelengths and SERS enhancement can be finely tuned. This capability enabled the formation of excitation-wavelength-specific SERS hotspots in a controlled manner. Notably, these plasmene nanosheets exhibited superior SERS performance compared to commercial Klarite substrates, which are rigid, opaque, and unsuitable for direct chemical identification on banknotes or coins [107].

The fabrication of SERS wearable sensors can follow either top-down or bottom-up approaches, each offering distinct advantages and limitations [108]. Top-down methods, such as EBL, photolithography, and NIL, provide precise control over nanostructure size [93,109,110], shape, and arrangement, leading to highly reproducible and uniform SERS hotspots. These techniques are ideal for producing high-performance, reproducible sensors with optimized plasmonic properties. However, they are often costly, time-consuming, and require specialized equipment, making large-scale production challenging. In contrast, bottom-up approaches, such as chemical synthesis, self-assembly, and laser-induced nanostructuring, are cost-effective, scalable, and suitable for flexible and wearable applications. These methods allow for easy integration of plasmonic nanostructures onto flexible substrates, enhancing sensor adaptability for real-world applications like epidermal or textile-based sensors [111]. However, achieving uniformity and reproducibility in bottom-up approaches can be difficult, as nanoparticle aggregation and batch-to-batch variations may lead to inconsistent SERS enhancement. Ultimately, while top-down approaches ensure precision and reproducibility, bottom-up methods offer scalability and flexibility, making the choice of fabrication technique dependent on the specific requirements of the wearable SERS sensor [112].

### 3.2. Flexible Substrate Integration

#### 3.2.1. Materials for Flexibility (e.g., Polymers, Textiles)

Advancements in wireless communication have underscored the potential of compact, adaptable, highly efficient, and flexible meta-devices for various applications, including conformal designs, flexible antennas, and wearable sensors. The integration of plasmonic nanostructures onto flexible substrates requires materials that maintain mechanical flexibility while preserving the optical and electrical functionality of the nanostructures. Polymers such as PDMS (polydimethylsiloxane) [23], PET (polyethylene terephthalate) [113], and PI (polyimide) are commonly used due to their durability, transparency, and ease of processing.

Eccrine sweat pH serves as a valuable noninvasive biomarker for assessing various physiological conditions, while SERS enables rapid and highly sensitive optical detection of biomolecules in sweat. Chung et al. developed a flexible, nanofibrous SERS-active substrate using electrospun thermoplastic polyurethane (TPU) with gold sputter coating [114]. To enable precise sweat pH sensing, the substrate was functionalized with two widely used pH-sensitive molecules, 4-mercaptobenzoic acid (4-MBA) and 4-mercaptopyridine (4-MPy). The resulting sensor demonstrated high sensitivity, achieving pH resolution of 0.14 for 4-MBA and 0.51 for 4-MPy using just 1 μL of sweat. Notably, its performance remained stable over 35 days (*p* = 0.361). In addition to stability, the Au/TPU nanofibrous SERS sensors exhibited rapid sweat absorption, strong repeatability, and excellent reversibility. This fabrication approach offered a simple, scalable strategy for developing SERS-based sensors, with potential applications extending beyond sweat pH monitoring to broader health biomarker detection [114].

Beyond polymers, textile-based substrates are gaining interest for wearable applications, offering a combination of flexibility, breathability, and comfort [115,116,117]. Advances in material science have led to the advancement of hybrid substrates that integrate inorganic and organic components, providing enhanced mechanical resilience and thermal stability.

Furthermore, smart textiles can serve as a promising platform for wearable SERS plasmonic sensors, enabling real-time, non-invasive chemical and biological detection. By integrating plasmonic nanostructures into textile fibers, these fabrics can enhance Raman signals for highly sensitive molecular detection. Advanced fabrication techniques, including dip-coating, electrospinning, and laser-induced nanostructuring, allow for the uniform deposition of plasmonic materials onto flexible, breathable fabrics without compromising their mechanical properties [118,119,120]. Smart textiles embedded with SERS-active surfaces can be used for continuous monitoring of biomarkers in sweat, detection of environmental toxins, and real-time disease diagnostics. Additionally, their washability, durability, and compatibility with flexible electronics make them well-suited for integrated wearable sensing systems. By combining wireless communication and miniaturized Raman spectrometers, SERS-enabled smart textiles could revolutionize point-of-care diagnostics, military applications, and wearable health monitoring [121].

Liu et al. introduced a highly adaptable wearable SERS sensor featuring an ultrathin, stretchable, adhesive, and biointegrable gold nanomesh, designed for cost-effective and scalable fabrication [122]. Its customizable design allowed seamless integration onto various surfaces, enabling real-time, label-free detection of analytes across a wide concentration range (10 to 10^6^ × 10^−9^ M). To demonstrate its functionality, the sensor was tested for sweat biomarker analysis, drug screening, and microplastic detection. This advancement paved the way for practical, versatile, and widely accessible wearable sensing technology.

To demonstrate the versatility of the wearable SERS sensor, its intrinsic adhesivity on various surfaces was tested without additional adhesives. Figure 6a–c show its secure attachment to a wrist for sweat biomarker detection, maintaining flexibility and biocompatibility with no adverse effects. Figure 6d–e illustrate its use on a cheek and contact lens for tear biomarker analysis, while Figure 6f highlights its application on a face mask for detecting breath and saliva biomarkers, relevant to COVID-19 and respiratory diseases. Figure 6g–i showcase its role in environmental monitoring, adhered to an elevator panel, door handle, doorknob, and keyboard, with drying options available for non-water-resistant surfaces. Additionally, Figure 6i,k demonstrate its use in food safety, applied to an apple and a leaf. To validate its practical utility, sweat biomarker detection, drug identification, and microplastic analysis were explored [122].

Recently, Rao et al. presented a highly durable, cost-effective (~10 cents per mm^2^), and flexible SERS substrate designed for real-world applications [123]. The ultrathin, adhesive nanomesh featured a three-dimensional network of silver-coated PVA nanofibers, ensuring a high density of plasmonic hotspots and reliable signal reproducibility (<20% CV). A protective PVA layer preserves the silver from oxidation, extending its usability beyond one month. Furthermore, the substrate supported a broad range of excitation wavelengths—488, 532, 633, and 785 nm—covering both visible and near-infrared regions. Demonstrating exceptional versatility, this substrate was successfully applied in food safety testing by detecting pesticide residues on fruit and in health monitoring through the analysis of human biofluids such as sweat and urine. This innovation represented a significant step toward transitioning SERS technology from controlled laboratory settings to industrial and real-world applications [123].

The silver/PVA nanomesh fabrication relies on long PVA nanofibers (>100 μm) as a template (Figure 7a). Electrospinning an 8 wt% PVA solution produced continuous fibers, which self-assembled into networks on a drum lined with cooking paper. A 300 nm-thick silver layer was then deposited via thermal evaporation (2 × 10^−4^ Pa, 0.11 nm s^−1^). Due to the unidirectional deposition, silver primarily coated one side, forming a concave-profiled nanomesh. The silver-coated PVA fibers remained flexible and durable for easy transfer onto various substrates. Finally, a water spray dissolved the PVA template, leaving behind a clean, flexible silver nanomesh. Cost analysis showed material and processing expenses, including PVA (12 cents/g), silver granules (700 cents/g), electrospinning (700 cents/day), and thermal evaporation ($3.6/day). This enabled large-scale fabrication (15 cm × 15 cm) at <10 cents per 1 mm^2^ (excluding labor), making SERS more accessible [123].

Structural analysis (Figure 7b–d) confirmed a thickness below 10 μm (Mitutoyo 547-401). SEM imaging (Regulus 8230, Hitachi) revealed an interconnected silver fiber network (>100 μm length, ~400 nm diameter), with fused junctions enhancing structural integrity. A bottom-view SEM image (Figure 7e) confirmed complete PVA removal, showing solid silver ribbons (~80 nm thick) with concave cross-sections and embedded silver nanostructures (~30 nm) [123].

Moreover, ensuring the reusability of SERS substrates is crucial for continuous, in situ monitoring applications, such as wearable sensors for biomarker detection in sweat. The challenge lies in effectively regenerating the SERS substrate after analytes have absorbed onto it, as desorption can be difficult, hindering long-term monitoring. Several strategies have been explored to address this issue. In situ electrochemical regeneration has been demonstrated as an effective method to refresh SERS substrates. A study proposed by Sibug-Torres et al. detailed a process where an oxidizing potential of +1.5 V (vs. Ag/AgCl) was applied for 10 s to remove adsorbates from Au nanoparticle monolayers. This was followed by a reducing potential of −0.80 V for 5 s in the presence of a stabilizing molecule, cucurbituril, to regenerate the nanogap hotspots. This method allowed the substrates to be reused over multiple cycles with minimal variation in performance [124].

Integrating self-cleaning materials into SERS substrates can facilitate the removal of adsorbed analytes. For instance, a porous TiO_2_-Ag core-shell nanocomposite material has been developed, exhibiting efficient self-cleaning properties. The TiO_2_ component, known for its photocatalytic activity, can degrade organic contaminants upon exposure to light, thereby renewing the SERS-active surface [125].

Microfluidic systems offer precise control over fluid flow, enabling the introduction of cleaning agents or fresh sensing surfaces to the SERS substrate. A study conducted by Sun et al. described an optofluidic SERS chip that utilized UV light to reduce a silver nitrate solution, forming Ag nanoparticles on a titanium dioxide-coated carbon fiber cloth. This setup allowed for online fabrication, molecular detection, and self-cleaning, demonstrating the potential of microfluidic-assisted approaches in maintaining SERS substrate performance [126]. Incorporating these strategies can enhance the longevity and reliability of SERS substrates in continuous monitoring applications, addressing the challenges associated with analyte desorption and substrate regeneration.

#### 3.2.2. Methods of Attaching Plasmonic Nanostructures to Wearable Substrates

Achieving robust attachment of plasmonic nanostructures to flexible substrates is critical for maintaining device performance under mechanical deformation [127]. Techniques such as thermal annealing, laser sintering, and plasma treatment are employed to enhance adhesion at the nanostructure–substrate interface [128,129,130,131]. Alternatively, printing techniques like inkjet or screen printing can directly deposit plasmonic nanoparticles onto flexible substrates with high precision [132,133]. Encapsulation layers and surface treatments are often applied to protect the nanostructures from environmental degradation, ensuring long-term stability and functionality [134,135].

Wang et al. presented tape nanolithography, a rapid, cost-effective method for fabricating flexible nanophotonic devices [136]. It involved soft lithography to create air-void nanopatterns on a donor substrate, followed by material deposition and direct transfer using adhesive tape. This technique achieved sub-100 nm resolution over large areas while allowing precise control over pattern shape, size, and material composition. After transfer, the residual material on the donor substrate formed complementary nanostructures, retaining functionality. The method enabled tape-supported plasmonic, dielectric, and metallo-dielectric nanostructures for applications in refractive index sensing, conformable plasmonics, and Fabry–Pérot cavity resonators (Figure 8). Additionally, it supported the transfer of freestanding plasmonic nanohole films onto unconventional surfaces, expanding its potential for advanced photonic integration.

### 3.3. Powering and Readout Mechanisms

#### 3.3.1. Miniaturization of Electronic Components

The miniaturization of electronic components is essential for integrating powering and readout mechanisms into compact, wearable plasmonic devices [47,137]. Advances in microfabrication have enabled the development of nanoscale batteries, supercapacitors, and energy harvesting systems that can provide sufficient power without compromising device flexibility [138]. For instance, thin-film batteries and flexible photovoltaic cells are being tailored for integration into wearable systems [139]. Moreover, the use of energy-efficient components, such as low-power microcontrollers and advanced signal processing units, reduces the overall power demand, enhancing the operational lifetime of wearable devices [140].

The development of fully functional wearable SERS sensors requires the seamless integration of three critical components: miniaturized excitation and detection systems, real-time data processing strategies, and continuous biomarker monitoring capabilities. Without these elements, wearable SERS technology remains limited to flexible or portable implementations rather than true wearables capable of autonomous, real-time analysis. Addressing these challenges is crucial for advancing wearable SERS sensors from laboratory research to practical, real-world applications [141].

One of the primary challenges in wearable SERS sensor development is the miniaturization of excitation and detection components. Traditional Raman spectroscopy systems rely on bulky laser sources and spectrometers, making them impractical for continuous, on-body monitoring. Recent advancements in micro-optics and photonic integration have enabled the development of compact Raman sources, such as miniaturized laser diodes and MEMS-based spectrometers [142,143]. However, integrating these elements into a flexible, low-power, and lightweight platform remains an ongoing research challenge. Addressing power consumption and optical alignment in a wearable form factor is essential to achieving a self-sufficient, miniaturized SERS system.

Another crucial aspect is the implementation of real-time data processing strategies [144]. Wearable SERS sensors generate complex spectral data that require sophisticated algorithms for noise reduction, peak identification, and biomarker quantification. Traditional Raman analysis relies on offline computation using powerful desktop systems, but for real-time monitoring, edge computing solutions such as embedded machine learning models and AI-driven spectral analysis are necessary. The integration of low-power microprocessors capable of executing real-time spectral analysis and wireless data transmission would enable wearable SERS sensors to function independently, making them more suitable for continuous health monitoring applications.

Finally, continuous biomarker monitoring is essential for wearable SERS sensors to provide meaningful health insights [145]. Unlike traditional one-time SERS measurements, continuous sensing requires stable, reproducible, and regenerable SERS substrates that can interact with biofluids such as sweat, saliva, or interstitial fluid over extended periods. Additionally, microfluidic integration could facilitate sample collection and transport to the active sensing area, enhancing sensor stability and performance. Ensuring that the sensor remains functional over time without significant signal degradation is key to achieving practical, long-term wearable SERS solutions.

Min et al. presented a self-sustaining wearable biosensor that operated using a perovskite solar cell, enabling continuous, non-invasive monitoring of metabolism [146]. The device featured a flexible, quasi-two-dimensional perovskite solar cell module capable of generating adequate power under both indoor and outdoor lighting. It achieved power conversion efficiency exceeding 31% under indoor illumination. The wearable system was shown to continuously collect various physicochemical data—including glucose, pH, sodium ion levels, sweat rate, and skin temperature—during physical activities, both indoors and outdoors, for more than 12 h [146].

A portable spectrometer holds significant value in various fields, offering real-time, on-site analysis of materials and substances without the need for bulky, stationary equipment. Its compact design enables rapid, non-destructive measurements in diverse environments, making it invaluable for applications in environmental monitoring [147], healthcare [148], agriculture [149], and food safety [150]. By providing precise data on light absorption, emission, or scattering, portable spectrometers help enhance decision-making processes, improve efficiency, and facilitate on-the-go research and diagnostics, all while reducing the costs and time associated with traditional laboratory-based methods. Li et al. introduced a flexible, wearable plasmonic sensor based on a paper-based microfluidic platform, utilizing SERS and paired with a portable Raman spectrometer for continuous, in situ sweat monitoring [151]. The device incorporated segmented channels that regulate fluid flow, ensuring rapid signal saturation. The sensor’s ability was demonstrated to enhance molecular permeation and enrichment, enabling precise SERS detection and quantitative analysis of key biomarkers such as uric acid concentration and pH levels in sweat. To address the challenges of real-world application, the device’s mechanical durability, optical performance, signal accuracy, long-term stability, and biocompatibility were thoroughly assessed, confirming its exceptional wearability and capability to detect sweat biomarkers reliably. Additionally, the sensor’s low production cost, outstanding reproducibility of fingerprint signals, and high sensitivity emphasize the potential of this microfluidic SERS paper for wearable sweat analysis [151].

The advancement of flexible SERS substrates has paved the way for rapid, on-site analysis. When integrated with compact Raman spectrometers, SERS technology holds significant potential for point-of-care testing (POCT) [152]. To enable effective in situ detection, it is crucial to develop high-performance, transparent, and flexible SERS substrates through a simple and cost-effective fabrication process. Over the past two decades, Raman spectrometers have undergone significant miniaturization while achieving improved performance. These advancements have been driven by enhanced integration of components, the adoption of transmission gratings instead of reflection gratings, and progress in electronics, display technology, and battery efficiency [153].

Zhang et al. introduced an efficient method for producing a large-area, flexible, and transparent SERS substrate. The design featured a silver-nanoparticle-grafted wrinkled polydimethylsiloxane (Ag NPs@W-PDMS) film, fabricated using a combination of surface-wrinkling techniques and magnetron sputtering [154]. When tested with rhodamine 6G as a probe molecule using a portable Raman spectrometer, the substrate exhibited a remarkable detection sensitivity (10^−7^ M), a high enhancement factor (6.11 × 10^6^), and strong reproducibility across different spots (9.0%) and batches (4.2%). Additionally, the Ag NPs@W-PDMS substrate retained its SERS efficiency even after 100 mechanical deformation cycles, including bending and twisting, and remains stable after 30 days of air exposure. To assess its practical applicability, in situ detection of malachite green on apple and tomato peels was successfully demonstrated, achieving a detection threshold of 10^−6^ M. Furthermore, to enhance its usability in POCT applications, a wireless transmission system was incorporated to instantly relay SERS spectral data to a computer for real-time processing and analysis. In summary, the Ag NPs@W-PDMS SERS substrate, developed through a scalable and straightforward fabrication method, in combination with portable Raman spectroscopy and wireless data transmission, presented a promising step toward transitioning SERS technology from laboratory research to real-world POCT applications.

When it comes to commercial products, BaySpec Inc. has introduced a groundbreaking watch-sized, wearable Raman spectrometer, setting a new standard for portable chemical analysis. Raman spectroscopy, a technique that identifies substances based on their molecular vibrations, has traditionally been confined to laboratory settings due to the size and complexity of the equipment. However, BaySpec has managed to miniaturize this technology into a device small enough to be worn on the wrist. This innovation allows users to conduct real-time, on-the-spot chemical identification without the need for bulky instruments, making spectroscopy more accessible and convenient [155]. The miniaturization of a Raman spectrometer into a wearable form is a significant engineering achievement. The device incorporates essential components such as a laser source, a spectrometer, and a detector into a compact, lightweight structure. Despite its small size, it retains high-performance capabilities, enabling accurate and reliable spectral analysis. Unlike traditional methods that require sending samples to a lab for testing, this device allows immediate substance identification. This level of portability is particularly beneficial for professionals who need instant results in dynamic environments, such as forensic investigators, pharmaceutical researchers, and field scientists.

One of the most important aspects of this wearable Raman spectrometer is its ability to provide non-destructive analysis. Traditional chemical identification methods often require altering or destroying samples, whereas Raman spectroscopy allows users to analyze substances without causing any damage. This feature makes it ideal for applications in pharmaceutical quality control, where verifying the composition of drugs without compromising their integrity is crucial. Law enforcement agencies can use it to detect illicit drugs, explosives, and hazardous materials during field operations. In food safety, it can help identify contaminants and ensure product authenticity, preventing fraud and health risks. Environmental scientists can also utilize this technology to analyze pollutants and unknown chemicals on site, reducing the need for extensive sample collection and transportation. The wearable spectrometer is expected to incorporate wireless connectivity, such as Bluetooth or Wi-Fi, to enable seamless data transfer to smartphones, tablets, or cloud-based systems. This connectivity allows users to store, analyze, and share their findings in real time, improving workflow efficiency. By integrating with AI and advanced data processing software, the device can automatically identify substances and provide insights, making Raman spectroscopy even more user-friendly and accessible. These smart features could significantly enhance decision-making in critical situations, such as detecting hazardous substances in public spaces or ensuring the authenticity of high-value pharmaceuticals.

This groundbreaking innovation has the potential to transform the field of portable spectroscopy. As industries continue to demand smaller, more efficient analytical tools, BaySpec’s wearable spectrometer could pave the way for a new generation of compact and intelligent spectroscopic devices. Beyond its current applications, future developments in this technology could lead to its use in medical diagnostics, personalized healthcare, and industrial quality control. By making advanced chemical analysis more accessible and mobile, BaySpec is pushing the boundaries of spectroscopy, ensuring that critical information is available whenever and wherever it is needed [155].

#### 3.3.2. Advances in Wireless Communication for Data Transmission

Wireless communication technologies are revolutionizing the readout mechanisms of wearable plasmonic devices by eliminating the need for bulky wired connections. Protocols such as Bluetooth Low Energy (BLE) [156], Near-Field Communication (NFC) [157,158], and advanced Wi-Fi standards enable seamless real-time data transmission between the wearable device and external readers or smartphones [159]. However, achieving efficient wireless communication while maintaining low power consumption presents several challenges. Bluetooth, NFC, and Wi-Fi are considered as potential communication methods, but each of these technologies comes with trade-offs. BLE is a strong candidate due to its minimal power consumption, but it has limitations in data transmission speed and range [160]. NFC, while energy-efficient, requires proximity, making it less suitable for continuous, real-time monitoring [161]. Wi-Fi offers higher data transfer rates and greater range but at the cost of significantly increased power consumption, which is a critical limitation for wearable devices operating on small batteries [162].

A key challenge in integrating these wireless communication methods into wearable SERS sensors is optimizing power efficiency without compromising real-time performance [163]. Efficient power management strategies, such as duty cycling, adaptive transmission rates, and low-power microcontrollers, can help extend battery life. Additionally, edge computing techniques can be implemented, where data preprocessing is performed on the device before transmission, reducing the amount of data sent wirelessly and consequently lowering energy consumption [164]. Emerging technologies like energy harvesting, which utilize body movement or ambient energy sources to power the device, could further enhance power efficiency. A hybrid approach that dynamically switches between communication modes based on data urgency and battery status could be an effective solution. Addressing these challenges will be essential for developing practical, long-lasting wearable SERS devices capable of reliable, real-time wireless monitoring [165].

Kang et al. presented an affordable and user-friendly healthcare platform designed to monitor heart rate and skin temperature using a wireless, battery-free sensor integrated with a tailored smartphone application. Kang et al. developed a custom NFC-enabled healthcare device and a companion smartphone app for seamless real-time data collection and analysis [157]. Figure 9a illustrates the exploded view of the wireless sensor, highlighting the composition of each layer. The device consisted of a dual-layer coil, circuitry, electronic components (including an NFC chip, LED, photodiode (PD), thermistor, resistors, and capacitors), and a black encapsulation layer. To achieve stronger magnetic power in a compact form, the dual-layer coil design allowed for more turns within the same dimensions. The layers of the coil were separated by a polyimide film and connected via a via-hole. Additionally, resonance frequency (*f*_res_) tuning was conducted to ensure effective communication with the NFC reader. As shown in Figure 9b, the dual-layer device exhibited greater attenuation, and a more defined resonance curve compared to the single-layer design. Consequently, the dual-layer device demonstrated a higher Q-factor, as seen in Figure 9c. As shown in Figure 9d, the device was encased in a black elastomer to block direct light from the LED to the PD. This layer also protected the circuit from degradation and skin contact, preventing leakage current. The LED operated at a safe current of ~1.66 mA, verified by a parameter analyzer, and the material safety data confirm it is safe for humans. A biocompatible double-sided tape ensures secure skin adhesion, confirming the device’s safety from both electrical and material standpoints. The quarter coin illustrated the device’s compact size. As shown in Figure 9e, the device includes an NFC chip and coil for wireless operation, driving the LED, and reading data from the PD and thermistor for biological data collection. This innovation offered a practical and cost-effective solution for advancing wireless wearable healthcare technologies while enabling effortless transmission of health data to professionals for remote, personalized consultations.

Recent developments in ultra-wideband (UWB) and mmWave communication offer higher data rates and lower latency, facilitating complex data processing for applications like health monitoring and environmental sensing. The integration of antenna systems directly into flexible substrates further enhances the usability and functionality of wireless-enabled plasmonic devices [166]. In [167], Jian utilized control variable methods, comparative analysis, and experimental approaches to explore several key performance aspects, including step counting accuracy, heart rate measurement precision under varying exercise conditions, data transmission speeds across different wireless technologies, and the performance of wearable devices in extreme environments (e.g., high and low temperatures, waterproofing, and battery life). The findings revealed that the wearable device tested demonstrated step accuracy above 96.5% and heart rate measurement accuracy exceeding 92% across a range of physical activities. Data transmission was fast and stable, and the device excelled in all other performance tests. Wearable intelligent devices, driven by wireless networks, hold immense potential for improving quality of life, health management, and smart services. As technology continues to evolve and user needs shift, these devices will likely advance further, offering even greater convenience and intelligence in everyday life.

## 4. Recent Advances in Wearable Plasmonic Sweat Sensors Based on SERS Technique

Advances in integrated circuits and wireless communication technologies have paved the way for electrochemical epidermal sensors that convert chemical signals into electrical outputs. These systems predominantly utilize amperometric [168], potentiometric [169], and voltammetric techniques [170] for analysis. In contrast to electrochemical sensors, which require complex circuitry and battery support, optical devices present a streamlined and compact alternative for advancing sweat analysis technologies. Techniques such as colorimetry [171], SERS [51], fluorescence [172], and electrochemiluminescence (ECL) [173] are commonly employed in wearable optical devices for sweat analysis (Figure 10). By utilizing chromophore or fluorophore molecules, these devices translate chemical signals in sweat into optical outputs, which can be interpreted directly or read using external devices such as smartphones or portable Raman spectrometers. The fusion of optical sensing with state-of-the-art flexible electronics and microfluidic systems has significantly accelerated the development of versatile platforms for efficient sweat collection and analysis [174]. In this section, we focused on wearable plasmonic sensors utilizing the SERS technique.

Wearable sweat sensors are increasingly recognized for their ability to provide valuable insights into health and disease conditions [28]. Conventional designs often rely on enzymes and antibodies to detect specific metabolites and stress biomarkers in sweat. However, these biological elements can degrade over time, reducing the effectiveness of the sensors. To overcome this limitation, a novel wearable system using plasmonic paper-based microfluidics was developed, enabling continuous, simultaneous analysis of sweat parameters such as volume, flow rate, and chemical composition [29]. By leveraging SERS, this system achieved highly sensitive, label-free detection of analytes, offering a precise chemical “fingerprint” for identifying substances like uric acid at both normal and abnormal levels. The incorporation of paper microfluidics ensured controlled sweat flow, allowing accurate monitoring of sweat dynamics. The device’s flexible and stretchable design ensures a secure and comfortable fit on the skin, minimizing the risk of irritation and making it ideal for extended use. This innovation paved the way for more reliable, non-invasive monitoring solutions, advancing personalized health care and disease management.

Figure 11a shows a flexible and lightweight plasmonic device worn on the wrist to collect, channel, store, and analyze sweat in real time using a portable Raman spectrometer. The system is composed of several layers, including an adhesive backing, a layer to block laser interference, a microfluidic channel made from paper, plasmonic sensing elements, and a protective encapsulation layer (Figure 11b,c) [29]. The serpentine design, crafted from cellulose, promotes efficient sweat flow through capillary action while ensuring flexibility and durability to withstand skin movements. A stretchable adhesive layer secures the device to the skin, and a carefully sized inlet, matching the width of the channel, optimizes sweat collection while minimizing inconsistencies in measurements. Encapsulation isolates the sweat within the device, protecting it from contamination. A similarly sized outlet prevents pressure buildup within the channel system.

Plasmonic sensors positioned along the channel quantify analytes over time using Raman spectroscopy. A black carbon adhesive layer blocked the laser and protected skin during measurements. The transparent PDMS encapsulation layer offered optical clarity, defined Raman bands for analyte reference, and protection against sweat evaporation and contamination. Figure 11d illustrates an image of the completed device, which includes an inlet on the encapsulation side. Gold nanorods (AuNRs) were produced via a seed-mediated approach. To maximize the enhancement of the SERS signal, the LSPR of the AuNRs was adjusted to coincide with the excitation wavelength of the laser. AuNRs with an LSPR peak at 765 nm were selected to match the 780 or 785 nm wavelengths typically used in both bench-top and portable spectrometers [29].

AuNRs were affixed onto the paper substrate through weak interactions, such as electrostatic forces and van der Waals forces. The produced AuNRs displayed a consistent size distribution, measuring 56.9 ± 3.6 nm in length and 14.5 ± 1.5 nm in diameter (Figure 11e). Optical extinction analysis revealed two plasmonic peaks at 511.5 nm and 765.0 nm, corresponding to the transverse and longitudinal localized surface plasmon resonances (LSPR) of the AuNRs (Figure 11f). When immobilized on the paper, the longitudinal LSPR band experienced a 48.3-nm blue shift due to changes in the surrounding RI. The extinction spectrum of the AuNR paper remained consistent with the solution, indicating a uniform AuNR distribution on the paper. SEM images confirmed this uniformity, with images showing AuNRs evenly distributed across the heterogeneous cellulose fiber surface (Figure 11g). This uniform distribution was critical for producing consistent Raman signals from the SERS substrates (Figure 11h,i) [29].

Furthermore, He et al. presented a novel wearable microfluidic nanoplasmonic sensor designed to perform portable and refreshable molecular analysis of critical sweat biomarkers, including urea, lactate, and pH [175]. The device featured a compact plasmonic metasurface engineered with uniformly distributed mushroom-shaped nanostructures, which provided enhanced sensitivity through high-performance SERS. Integrated into a microfluidic platform, this system addressed the challenges of traditional wearable SERS devices, such as contamination from mixed sweat samples, by enabling controlled and precise sweat flow. This allowed for high temporal resolution and repeatable measurements. The sensor was paired with a lightweight, customized Raman analyzer featuring an intuitive interface, facilitating real-time detection and analysis of sweat biomarker signatures. By combining microfluidic technology with portable SERS capabilities, this dynamic and user-friendly device offered a powerful solution for personalized health management and continuous biofluid monitoring [175].

Xiao et al. introduced a non-invasive, microfluidic-based wearable plasmonic sensor designed for simultaneous sweat sampling and real-time monitoring of acetaminophen levels as a vital sign indicator [23]. At the heart of the sensor was an array of Au nanosphere cones, which served as the sensing platform with enhanced SERS activity. This design enabled highly sensitive, non-invasive detection of acetaminophen by capturing its unique SERS spectral fingerprint. The sensor exhibited exceptional sensitivity, accurately detecting and quantifying acetaminophen concentrations as low as 0.13 μM. The sensor’s performance was further accessed when integrated with a Raman spectrometer to monitor acetaminophen levels in subjects who had received the drug. The results demonstrated the sensor’s ability to precisely measure acetaminophen and provided valuable insights into drug metabolism. This innovative sweat sensor technology represented a significant leap in wearable sensing, offering label-free, highly sensitive molecular tracking for non-invasive, point-of-care drug monitoring and personalized management [23].

Recently, Atta et al. introduced a groundbreaking wearable patch that utilized SERS for the concurrent detection of three vital sweat biomarkers: lactate, urea, and glucose [20]. The patch was designed with sharp-branched gold nanostars (GNS) that exhibited high plasmonic activity and are fabricated onto a commercially available adhesive tape. This approach ensures a flexible, durable, and cost-efficient solution. Figure 12a illustrates the preparation process for a multibranched GNS wearable patch, which involves a simple method of applying highly concentrated GNS onto an adhesive tape. The adhesive tape’s pronounced hydrophobicity causes the GNS to concentrate in a localized area. After the GNS solution was applied, it was left to air dry. Four different concentrations of GNS were tested to fabricate the wearable patches. The GNS solution was concentrated to 2x, 5x, and 10x the original concentration, resulting in wearable patches WP-1, WP-2, WP-3, and WP-4, respectively. These patches corresponded to the original, 2x, 5x, and 10x concentrated GNS solutions. The image of WP-4 (Figure 12b) demonstrates the GNS concentrated in a small region on the adhesive tape. Figure 12c shows SEM images of WP-4, revealing a high concentration and uniform distribution of GNS on the tape. Magnified SEM images (Figure 12d,e) further confirm that the GNS maintained their original morphology.

The device achieved detection limits of 0.7 μM for lactate, 0.6 μM for urea, and 0.7 μM for glucose, which were far below the clinically relevant concentrations typically observed in sweat. The patch’s effectiveness was thoroughly assessed during various physical activities, such as sitting, walking, and running, demonstrating its ability to track changes in biomarker levels in real time under dynamic conditions. By enabling accurate, simultaneous biomarker monitoring, this wearable sensor represented a significant step forward in noninvasive healthcare technology. Its affordability, simplicity, and real-time functionality made it an ideal tool for autonomous health tracking and home-based medical diagnostics, heralding a new era of personalized healthcare solutions [20].

Scalability is a significant factor in the development of wearable sensors based on SERS technique, determining their potential for mass production and widespread adoption. For these devices to transition from niche applications to broader markets, they must utilize cost-effective, easily reproducible fabrication methods while maintaining high performance and reliability. Despite their outstanding performance, these sensors face barriers to widespread adoption due to intricate manufacturing processes and limited multifunctional capabilities, restricting their utility to niche health monitoring applications. To address these shortcomings, Liu et al. unveiled a scalable, wearable SERS sensor crafted from an economical, ultrathin, flexible, stretchable, adhesive, and biointegratable Au nanomesh (Figure 13a,b) [122].

This cutting-edge sensor was straightforward to produce, adaptable in design, and suitable for application on various surfaces, facilitating label-free, extensive, on-site detection of a wide array of analytes across an expansive concentration range. This innovation marked a pivotal step forward in advancing wearable sensing technologies, enhancing their practicality, affordability, and widespread utility. Liu’s sensor design drew inspiration from a recently developed lightweight, stretchable electronic device, which was both inflammation-free and gas-permeable. Constructed using Au-coated biocompatible polyvinyl alcohol (PVA) nanofiber, it was designed to hold to human skin or uneven, flexible surfaces for extended periods [176]. Its optical properties, previously unexplored, were leveraged, and its structure was fine-tuned to enable SERS functionality while retaining key advantages such as ease of fabrication, affordability, ultra-thinness, flexibility, stretchability, strong adhesion, and biointegration (Figure 13c–f). The sensor, capable of being molded into various shapes, was suitable for use on almost any surface, providing label-free, large-scale, on-site detection of analytes over a broad concentration range ((10–10^6^) × 10^−9^ M). Its practicality was validated through successful recognition of sweat biomarkers, illicit substances, and microplastics [122].

## 5. Advancing Wearable Sensors Through AI

The integration of AI and ML in the development of wearable sensors is revolutionizing the field of biosensing and health monitoring [177]. AI and ML algorithms are being employed to enhance the functionality, accuracy, and user-friendliness of these sensors. AI models can process vast amounts of sensor data, identify patterns, and make predictions with minimal human intervention, greatly improving the decision-making process in clinical settings [178]. ML, a subset of AI, is particularly useful in training algorithms to adapt and learn from new data, allowing the wearable devices to optimize their performance over time [179]. For example, ML techniques such as supervised learning can be used to correlate sensor data with specific health conditions, making the wearable sensor capable of diagnosing or tracking the progress of diseases like diabetes, cardiovascular conditions, or even neurological disorders [180].

AI and ML also enable the real-time analysis of the signals produced by plasmonic sensors, which often involve complex data patterns due to the sensitivity of the nanosensors [181]. ML algorithms can be trained to differentiate between true positive signals and noise, improving the reliability of the sensor in dynamic and real-world environments. For instance, ML algorithms like deep learning (DL) can be used to improve signal processing in wearable plasmonic sensors by learning from large datasets of sensor responses and environmental variables [182]. This leads to better accuracy in detecting low-concentration biomarkers in the presence of background interference. Furthermore, AI can assist in predictive modeling, where the sensor data are continuously analyzed to forecast potential health issues before they become clinically evident, allowing for proactive health management [183].

AI-driven algorithms, such as ML and DL models, enhance spectral interpretation by reducing noise, identifying subtle spectral variations, and enabling rapid, automated biomarker recognition. These capabilities are crucial for field applications where real-time analysis is required without extensive preprocessing. For instance, convolutional neural networks (CNNs) have been applied to SERS data for accurate identification of biomarkers in medical diagnostics, demonstrating improved sensitivity and specificity [184]. Additionally, DL-assisted SERS has been utilized for rapid and direct nucleic acid amplification and detection, highlighting its potential in molecular diagnostics [185]. Furthermore, combining acoustic bioprinting with AI-assisted Raman spectroscopy has enabled high-throughput identification of bacteria in complex samples like blood, underscoring the practicality of AI-enhanced SERS in real-world scenarios [186]. By leveraging AI, the accuracy and efficiency of on-site Raman detection can be significantly enhanced, paving the way for broader real-world applications [187,188].

In addition to enhancing sensor performance, AI and ML are pivotal in advancing the miniaturization and user experience of wearable plasmonic sensors [189]. AI can streamline sensor design by optimizing parameters like material properties, sensor configuration, and detection techniques. In wearable technology, comfort and ease of use are key; AI can help design adaptive systems that adjust the sensor’s operation based on the wearer’s activity, environmental conditions, or health needs [190]. By employing AI-driven models, wearable sensors can become more energy-efficient, reducing the power consumption typically associated with continuous monitoring and making them more suitable for long-term, real-world use. Moreover, the integration of AI and ML into wearable sensors facilitates personalized medicine. ML can account for individual variability by tailoring sensor performance to specific users, considering factors such as age, gender, lifestyle, and pre-existing conditions. This ensures that the sensor provides accurate, relevant health data for each person, making the technology more effective in detecting early signs of diseases or managing chronic conditions. Overall, the role of AI and ML in wearable plasmonic sensors not only enhances their technical performance but also holds the potential to transform healthcare by enabling more accessible, efficient, and personalized health monitoring solutions [182].

Early detection of pathological markers is vital for reducing viral transmission, minimizing infection risks, and enhancing disease recovery rates. As a result, researchers worldwide are actively developing methods to identify pathological markers and environmental toxins [191]. However, conventional clinical techniques often rely on expensive, high-precision instruments and complex procedures, highlighting the urgent need for faster, simpler, and more cost-effective analytical solutions. Electrochemical biosensors have emerged as a promising alternative, offering advantages such as affordability, minimal sample preparation, rapid analysis, and ease of miniaturization and integration [192]. These qualities have made them a focal point of extensive research. Meanwhile, ML has gained prominence for its powerful data analysis and predictive capabilities, particularly in material synthesis and sensor design.

Bao et al. proposed a flexible wearable device featuring a ML-assisted carbon black–graphene oxide conjugate polymer (CB–GO/CP) electrode for portable tyrosine (Tyr) detection. Artificial neural networks (ANN) and support vector machines (SVM) were trained using feature data collected from artificial urine samples, alongside pH and temperature measurements [193]. As shown in Figure 14a, 5 g of ethyl cellulose, the primary hydrophobic agent, was gradually mixed into 100 mL of pine oil alcohol (organic solvent) under stirring at 60 °C until a clear, viscous colloid formed. A yellow oil-based dye was then added, producing a hydrophobic slurry with good flowability. A4-sized filter paper was placed beneath a 350-mesh screen and printed using a screen printer. The hydrophobic-patterned paper was dried at 60 °C for 30 min. Hydrophobic ink was then printed onto the paper using a custom stencil and cured at 120 °C for 30 min to form paper electrodes [193].

The preparation of carbon black–graphene oxide (CB–GO) ink involved separately stirring 1 mg of CB powder and 1 mL of GO dispersion in 10 mL solutions at 1000 rpm for one hour. Afterward, 3 mL of the CB solution was mixed with 4 mL of the GO dispersion, followed by 30 min of ultrasonication. The mixture was then combined with 70 mL of deionized water and 3 g of PVA, stirred at 120 °C for an hour, and processed in an ultrasonic cell mill for 20 min to produce the CB–GO ink. Screen printing was employed to layer materials on paper-based electrodes: carbon ink for the conductive pattern, Ag/AgCl ink for the reference electrode, and CB–GO ink for the active layers [193].

Figure 14b presents the hardware circuit and software interface of the portable device. A lithium battery, connected to two linear voltage regulator modules, maintained a steady 3.3 V output. The central component of the constant potential meter was a digital-to-analog converter (DAC) module controlled via an I2C interface. A polyimide (PI) substrate was used for the circuit board to ensure flexibility and portability. The electrodes were fixed in place with screws and nuts through pre-designed holes in the paper-based electrodes, as illustrated in Figure 14b. The device app, created using the WeChat app developer tool, was installed on a smartphone and paired with a low-power Bluetooth module for efficient data transfer [193].

## 6. Concluding Remarks and Outlook

In conclusion, wearable plasmonic sensors, particularly those based on surface-enhanced Raman spectroscopy (SERS), have emerged as a promising technology in the fields of health monitoring, environmental sensing, and human–machine interfaces, owing to their high sensitivity, real-time detection, and miniaturized form factors. SERS-based sensors offer remarkable improvements in analyte detection by enhancing Raman signals through the use of nanostructured materials, such as Au and Ag nanoparticles, significantly increasing sensitivity and selectivity. Recent advancements have significantly enhanced the performance and versatility of these sensors, with innovations in materials, design, and fabrication techniques enabling more efficient integration into wearable devices. Notable developments include the incorporation of these nanomaterials and the use of flexible substrates that allow for seamless integration into clothing and skin-contact devices. These sensors have found applications in continuous health monitoring (e.g., glucose, lactate, and sweat analysis), as well as in environmental sensing (e.g., detection of pollutants and allergens).

However, several challenges remain, particularly in terms of long-term stability, integration with wireless communication systems, and minimizing interference from external factors like motion and temperature fluctuations. Additionally, scaling up the production of these sensors while maintaining cost-effectiveness and performance remains a critical hurdle. To address these challenges, future research should focus on overcoming issues related to sensor durability and reliability in real-world environments, as well as advancing wireless communication technologies that enable seamless data transmission and integration with mobile health platforms.

Furthermore, expanding the range of detectable biomarkers and developing multifunctional sensors capable of monitoring various physiological parameters simultaneously will enhance the applicability of wearable plasmonic sensors in personalized medicine and proactive health management. Improving user-friendliness, including reducing sensor size and enhancing comfort, will be key to fostering widespread adoption of these devices. Future work should also explore novel energy-efficient solutions, such as energy harvesting from body movements or ambient sources, to extend sensor autonomy and reduce dependence on external power sources. As these technologies mature, interdisciplinary collaboration across fields such as nanotechnology, biotechnology, and wireless communications will be essential for realizing the full potential of SERS-based wearable plasmonic sensors in transforming healthcare, environmental monitoring, and human-computer interaction.

## Figures and Tables

**Figure 2 sensors-25-01367-f002:**
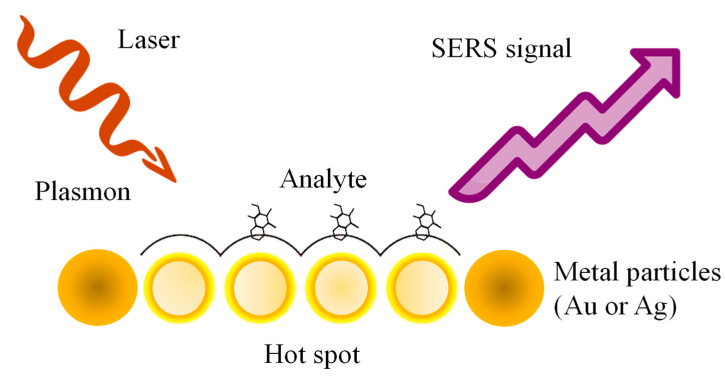
Schematization of SERS mechanism.

**Figure 3 sensors-25-01367-f003:**
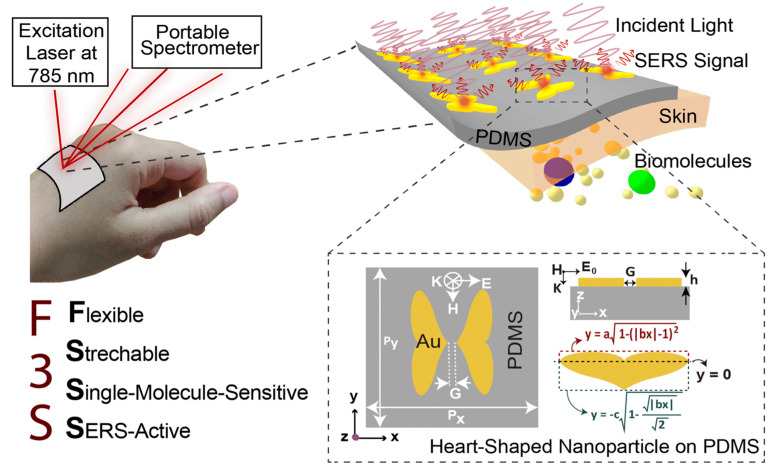
Schematic representation of the plasmonic SERS sensor, designed for direct application on the wrist to enable real-time, label-free sweat analysis using a portable Raman spectrometer. The **top inset** highlights a heart-shaped nanodimer array embedded in a flexible PDMS substrate for enhanced biochemical detection. The **bottom inset** illustrates the structural parameters of the heart-shaped nanoparticle (NP) [77].

**Figure 4 sensors-25-01367-f004:**
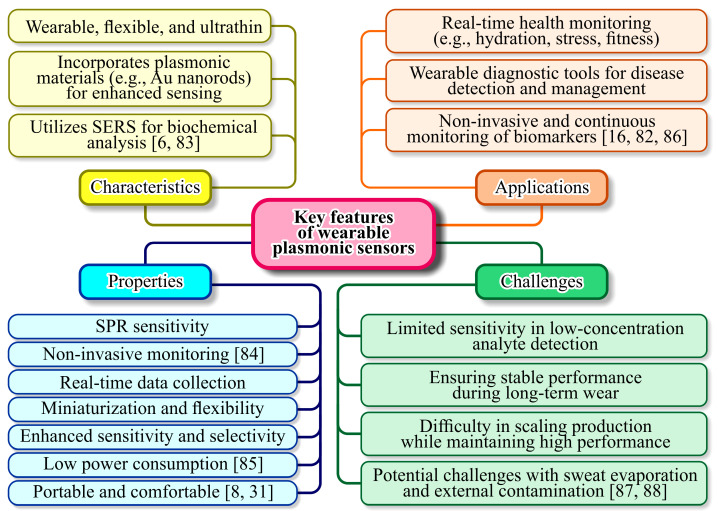
Category and details of key features of wearable plasmonic sensors [6,8,16,31,82,83,84,85,86,87,88].

**Figure 5 sensors-25-01367-f005:**
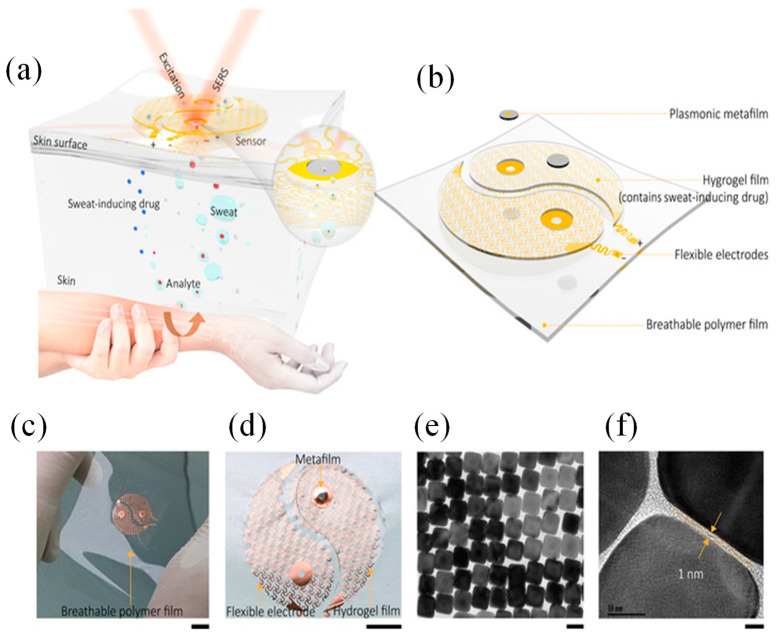
Wearable SERS device with plasmonic metamaterial [16]: (**a**) Schematic of the device’s working principle and design; (**b**) it features two main components: a sweat extraction module and a SERS sensing module, arranged in a yin-yang pattern. The inset highlights the sensing interface near the plasmonic metafilm. (**c**) Optical image of the device and (**d**) close-up of the sweat extraction module with a hydrogel layer (loaded with acetylcholine chloride) on the spiral mesh electrode to stimulate sweat glands; (**e**,**f**) TEM images of the plasmonic metafilm, formed by a Ag nanocube superlattice [16].

**Figure 6 sensors-25-01367-f006:**
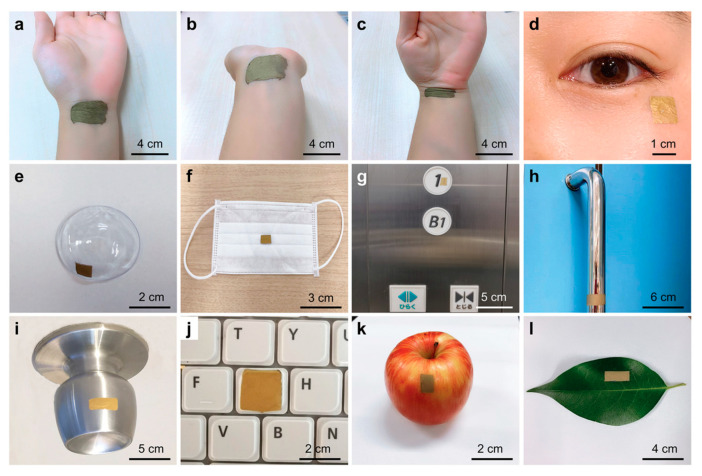
Images illustrating the wearable SERS sensor applied to various surfaces for multiple applications: (**a**–**c**) Sensor attached to a human wrist, retaining flexibility during movement. (**d**,**e**) Sensor placed on a cheek and contact lens for tear biomarker detection. (**f**–**j**) Sensor positioned on a face mask, elevator button, door handle, doorknob, and keyboard, showcasing its role in environmental and infection monitoring. (**k**,**l**) Sensor applied to an apple and leaf, demonstrating its potential in food safety assessment [122].

**Figure 7 sensors-25-01367-f007:**
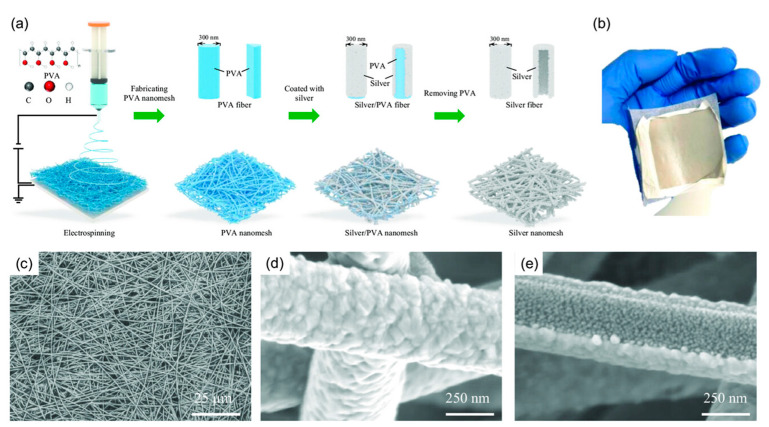
(**a**) Fabrication process: Electrospinning creates a PVA nanofiber template, followed by silver deposition via thermal evaporation. The PVA is then removed with a water spray. (**b**) Image of the final silver/PVA nanomesh. (**c**) SEM top view showing the overall structure. (**d**) Higher-magnification SEM image revealing fine details. (**e**) Bottom-view SEM image highlighting the concave structure and internal silver features [123].

**Figure 8 sensors-25-01367-f008:**
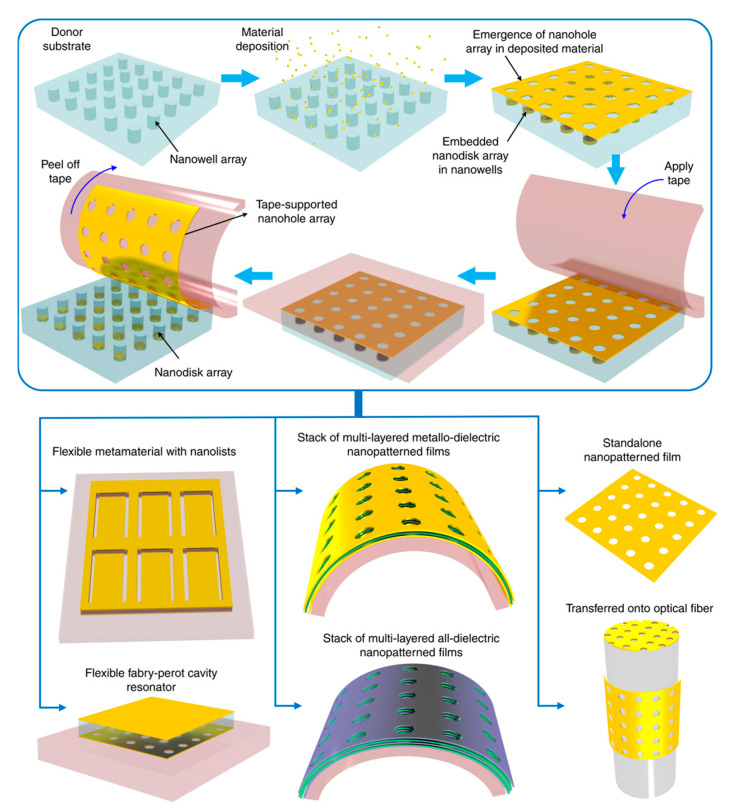
Tape nanolithography process for creating and transferring nanopatterned films onto adhesive tape and other surfaces [136].

**Figure 9 sensors-25-01367-f009:**
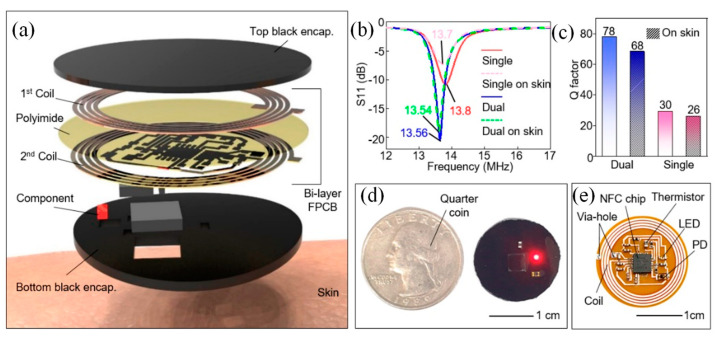
NFC-based wireless device [157]. (**a**) Exploded view of the device. (**b**) Frequency characteristics of single-layer and dual-layer devices on slide glass and skin. (**c**) Q-factor of dual-layer and single-layer devices on skin. (**d**) Image of the black-encapsulated device next to a quarter coin. (**e**) Photo of the unencapsulated wireless device [157].

**Figure 10 sensors-25-01367-f010:**
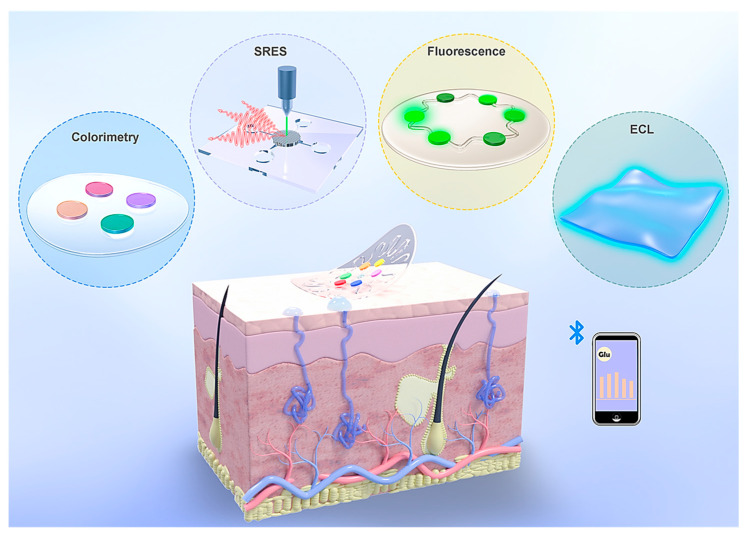
The four categories of optical sweat sensors are colorimetry, surface-enhanced Raman scattering (SERS), fluorescence, and electrochemiluminescence (ECL) sensors [174].

**Figure 11 sensors-25-01367-f011:**
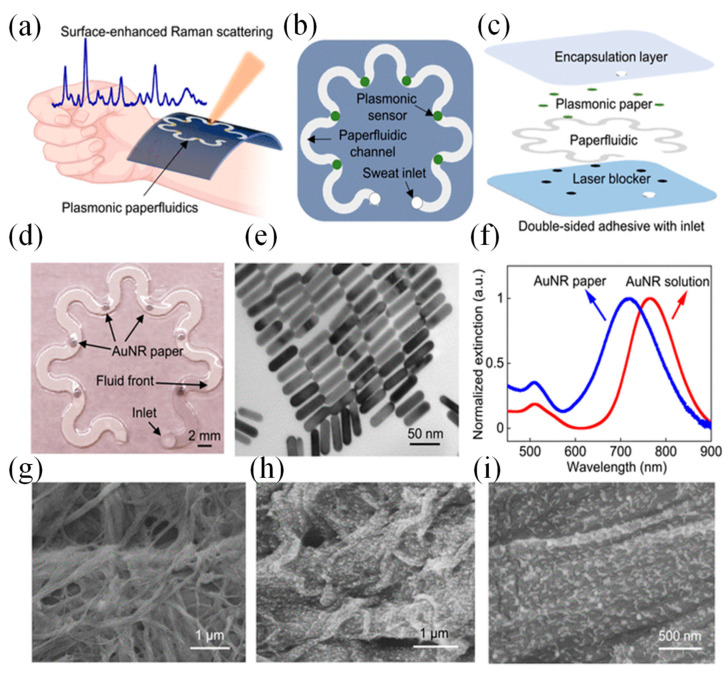
Wearable plasmonic paper microfluidic device [29]. (**a**) Illustration of the device for sweat collection, storage, and in situ SERS analysis. (**b**) Top and (**c**) side views showing the device’s functional layers. (**d**) Photo of the assembled device with six plasmonic sensors. (**e**) TEM image of AuNRs with uniform size. (**f**) Extinction spectra of AuNR solution and AuNR paper. (**g**) SEM image of plain chromatography paper, and (**h**,**i**) SEM images of AuNR-coated paper [29].

**Figure 12 sensors-25-01367-f012:**
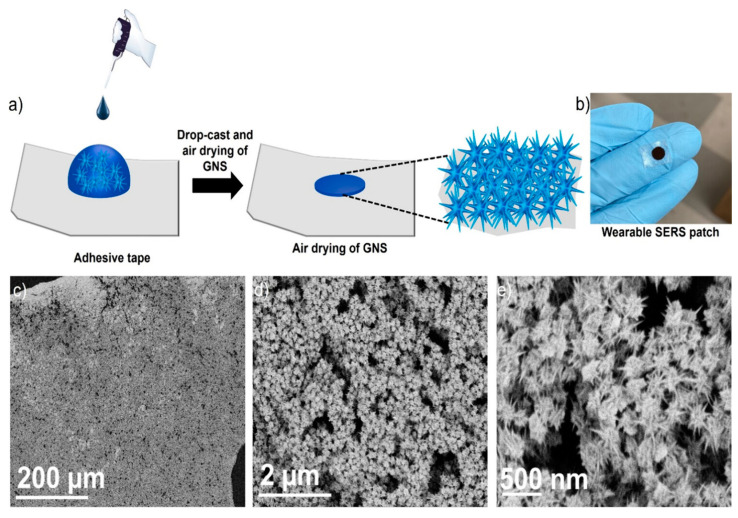
(**a**) Schematic depiction of the wearable patch preparation process. (**b**) Photograph of the GNS wearable patch. (**c**–**e**) SEM images of the wearable patch at different magnifications [20].

**Figure 13 sensors-25-01367-f013:**
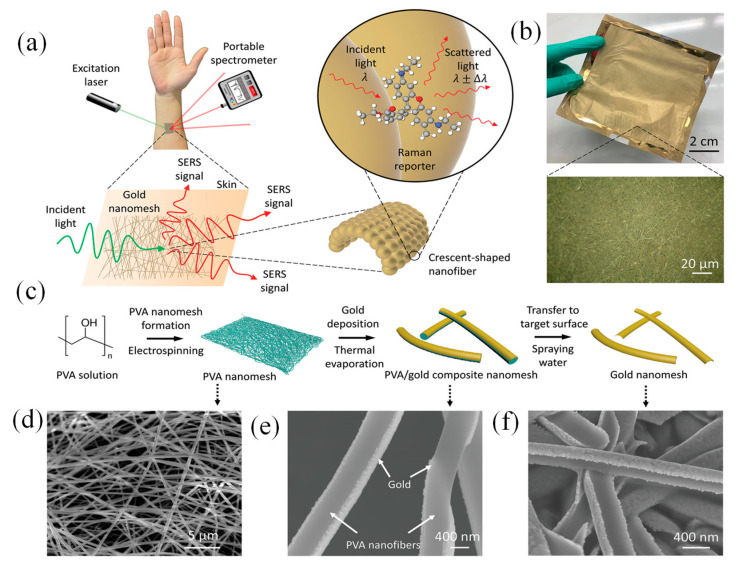
Overview of the wearable SERS sensor, highlighting its concept, fabrication, design, and characterization [122]. (**a**) Schematic representation of the wearable SERS sensor applied on the skin. (**b**) Photograph of the fabricated Au nanomesh, with the inset showing a 50× optical microscopy image detailing its structure. (**c**) Step-by-step fabrication process illustrated alongside corresponding SEM images. (**d**) PVA fiber nanomesh. (**e**) Au-coated PVA fiber nanomesh, and (**f**) the final Au nanomesh after the PVA fibers are removed [122].

**Figure 14 sensors-25-01367-f014:**
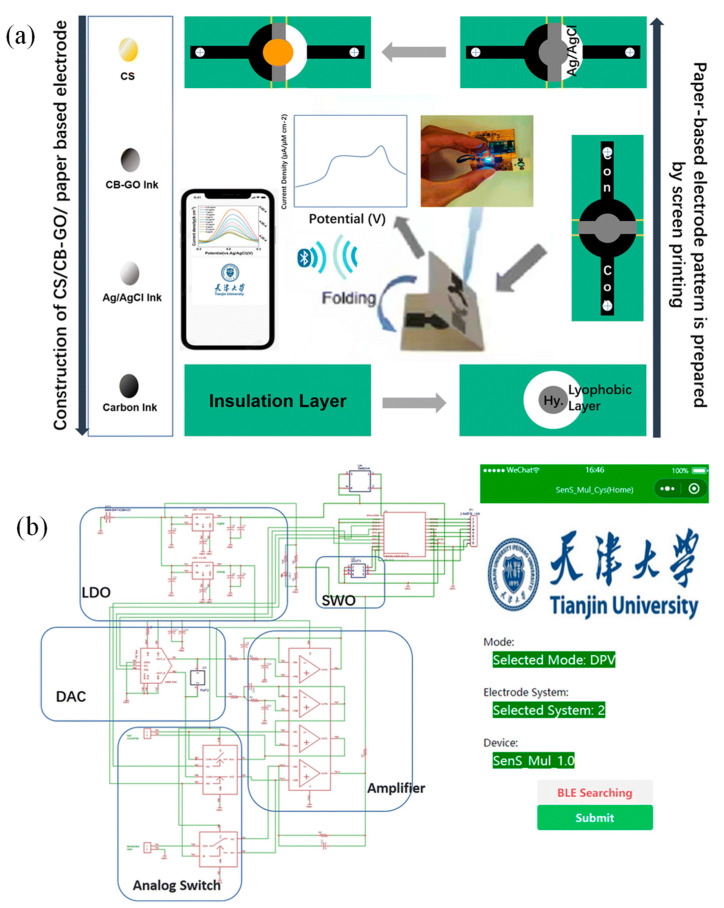
(**a**) Fabrication process for paper-based CS/CB–GO/CP electrodes designed for flexible device integration. (**b**) Circuit schematic and software interface for the flexible wearable immune sensor [193].

## Data Availability

Not applicable.

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
