# Peer review of "Trends and Advances in Wearable Plasmonic Sensors Utilizing Surface-Enhanced Raman Spectroscopy (SERS): A Comprehensive Review"

_sensors, 2025, doi:10.3390/s25051367_

Round 1

Reviewer 1 Report

Comments and Suggestions for Authors

1.“These sensors utilize the distinctive properties of Surface-Enhanced Raman Spectroscopy (SERS) to recognize a wide range of analytes with high sensitivity and selectivity.”

What are the distinctive properties of Surface-Enhanced Raman Spectroscopy (SERS)? This was not mentioned in the preceding text. Additionally, not all wearable sensors use SERS technology, so this statement needs to be revised.

2.“Despite their potential, scalable, cost-effective, and eco-friendly methods to fabricate high-performance SERS and SEPL substrates with broad plasmonic response and significant enhancement remain elusive.”

Please describe the fabrication methods of SERS and SEPL substrates prior to the single-pulse laser-induced silver-layer delamination used by Kuchmizhak et al. Also, explain why Kuchmizhak et al.'s method outperforms earlier methods in terms of performance and cost-effectiveness.

3.“When light illuminated the system, surface charges formed on the Au nanodisk, acting as dipoles and generating localized surface plasmon polaritons (LSPPs).”

What are localized surface plasmon polaritons (LSPPs)?

4.“Chang et al. developed a flexible LSPR biosensor by integrating metal–insulator–metal (MIM) nanodisks with a biocompatible PDMS substrate.”

Since Section 2.2 focuses on plasmonic materials, please describe the material innovations and improvements made in the development of LSPR biosensors by Chang et al. The discussion on the structural design of the sensor should be appropriately brief to avoid overshadowing the main focus on materials.

5. “To test the SERS activity, crystal violet (CV) was employed as an analyte on substrates with and without the nanoimprinted pattern.”

This example demonstrates an issue of imbalance in details. If this example is meant to introduce the top-down fabrication method, the focus should be on Colnita et al.’s use of nanoimprint lithography (NIL) to fabricate the sensor. Similar issues appear in other examples cited later in the text. Please revise these sections accordingly to maintain focus on the main topics. Additionally, appropriate citations should be added in this section.

6. In Section 3.2.2. Methods of Attaching Plasmonic Nanostructures to Wearable Substrates, citations should be added after the discussion of each method to enhance credibility.

7. “making it invaluable for applications in environmental monitoring, healthcare, agriculture, and food safety.”

Add citations corresponding to the applications mentioned in each field to strengthen credibility. Overall, the paper lacks sufficient and diverse references; additional citations and illustrative examples should be included.

8.This article presents an affordable and user-friendly healthcare platform designed to monitor heart rate and skin temperature using a wireless, battery-free sensor integrated with a tailored smartphone application.”

Which article is referred to as “this article”? There is no clear mention in the preceding or following text.

9.Additional examples can be included here, while simplifying the content of existing examples.

In summary, this paper requires major revisions

Comments on the Quality of English Language

No problems found in English expression

Author Response

Dear Reviewer, thank you for your valuable suggestions and constructive criticism. We have carefully revised the paper accordingly and appreciate your insightful feedback.

Additions and changes are marked by color in the manuscript.

Detailed answers are collected in the attached file.

We look forward to your positive response.

Reviewer 2 Report

Comments and Suggestions for Authors

See attached file.

Author Response

(The authors gave the same response as above.)

Reviewer 3 Report

Comments and Suggestions for Authors

In this work, Khonina and Kazanskiy reviewed the advancement of wearable SERS sensors and the corresponding design, fabrication and integration. In addition, the challenges, e.g., miniaturization, power consumption and stability were highlighted. The perspectives of next-generation wearable SERS sensing technologies for real-time health diagnostics were predicted. After carefully reading, I do not recommend to publish the manuscript in the present version. There some comments should be concerned by the authors as following:

(1) As highlighted in the title and abstract, the wearable SRES sensing technologies should be focused in the review, rather than plasmonic sensors, e.g., RI. They are different although the mechanisms are somewhat similar. As discussed from the last paragraph in the page 5 to page 8, the flexible LSPR biosensors integrating MIM were commonly used in LSPR sensing. The main types of nanostructure design for flexible SERS techniques should be reviewed.

(2) Table 1 could be replaced with a diagram to provide a clearer and more objective presentation.

(3) In Figure 3, the role of electrode should be explained in details. It is well-known that SERS is an optical technique, so it is insensitive to the static electrical field. The major contribution of this literature to flexible SERS detection should thereby be highlighted clearly.

(4) In the lines from 315 to 325, the authors introduced a supramolecular film sensor. Is this sensor related to SERS? I suggest the authors retaining the literatures closely relative with SERS, whereas the other plasmonic sensors should be removed. The studies on flexible SERS actually are sufficient to support the topic of the manuscript.

(5) The description in line 337-346 should be removed based on the same reason as mentioned above.

(6) Figure 4 is either not related to SERS.

(7) For section 3.1, the description of major advantages and drawbacks for the fabrication techniques should be provided. Additional, it would be beneficial to include relevant figures in this section from the literatures to make the discussion visually informative.

(8) The section 3.2.2 is nonspecific. The development should be discussed in details.

(9) For section 3.3, in my point of view, it is not the major challenge to SERS. The SERS detection system generally comprises three components, i.e., narrow-linewidth laser source, SERS substrate and Raman spectrometer. The laser source and spectrometer can be integrated in one platform, for which the handheld spectrometer is most compactable so far. The SERS substrate can be flexible. The on-chip laser source and spectrometer system for Raman detection, in fact, is too challenging. The design of high-performance SERS substrate is the most studied. The flexible SERS substrates should be discussed in this review. The general challenge in flexible plasmonic sensors is out of the scope of this manuscript.

(10) In section 5, the AI-promoted SERS analysis should be reviewed. This is very important for on-site Raman detection.

In my opinion, this manuscript should be further revised to provide deeper understanding of strengths and weaknesses of the wearable SERS biosensors, highlighting their limitations and giving insights into potential applications of SERS in future. I recommend a review published in 2019 for the similar topic, which would be useful to authors for reorganization of their manuscript. (Toward Flexible Surface-Enhanced Raman Scattering (SERS) Sensors for Point-of-Care Diagnostics. Advanced Science, 2019, 1900925)

Author Response

(The authors gave the same response as above.)

Round 2

Reviewer 1 Report

Comments and Suggestions for Authors

The revised article has met the publication requirements and is recommended for publication

Comments on the Quality of English Language

English terms are correct and readable

Author Response

Comment: The revised article has met the publication requirements and is recommended for publication.

Reply: Thank you for accepting our paper in its current form.

Reviewer 2 Report

Comments and Suggestions for Authors

Although the authors attempted to address some of the concerns I raised, their responses were not sufficiently comprehensive, particularly regarding the fabrication of flexible SERS substrates and the miniaturization of the Raman system. I encourage the authors to provide a more in-depth discussion rather than a superficial treatment of these topics. For instance, BaySpec Inc. has developed a watch-sized, wearable Raman spectrometer, which could serve as a relevant example for miniaturization efforts. I also suggested a few other literatures for authors to consider.

Author Response

Comment: Although the authors attempted to address some of the concerns I raised, their responses were not sufficiently comprehensive, particularly regarding the fabrication of flexible SERS substrates and the miniaturization of the Raman system. I encourage the authors to provide a more in-depth discussion rather than a superficial treatment of these topics. For instance, BaySpec Inc. has developed a watch-sized, wearable Raman spectrometer, which could serve as a relevant example for miniaturization efforts. I also suggested a few other literatures for authors to consider.

Reply: As suggested by the reviewer, we have provided some detail on wearable Raman spectrometer by BaySpec Inc.

Reviewer 3 Report

Comments and Suggestions for Authors

The revised manuscript is much better. It can be accepted as is. 

Author Response

Comment: The revised manuscript is much better. It can be accepted as is.

Reply: Thank you for accepting our paper in its current form.